# Central Line Associated Bloodstream Infections in Critical Ill Patients during and before the COVID-19 Pandemic

**DOI:** 10.3390/healthcare11172415

**Published:** 2023-08-29

**Authors:** Sona Hlinkova, Eva Moraucikova, Anna Lesnakova, Agnieszka Strzelecka, Vladimir Littva

**Affiliations:** 1Faculty of Health, Catholic University in Ružomberok, Námestie Andreja Hlinku 48, 034 01 Ružomberok, Slovakia; eva.moraucikova@ku.sk (E.M.); anna.lesnakova@ku.sk (A.L.); vladimir.littva@ku.sk (V.L.); 2Institute of Health Sciences, Collegium Medicum, Kochanowski University, Al. IX Wieków Kielc 19A, 25-317 Kielce, Poland; agnieszka.strzelecka@ujk.edu.pl

**Keywords:** central line-associated bloodstream infection, COVID-19, risk factors, device associated infection, health care-associated infection, surveillance

## Abstract

(1) Background: The purpose of this study was to evaluate the impact of the COVID-19 pandemic on the rates of central line-associated bloodstream infections (CLABSI), its etiology, and risk factors in critically ill patients, because Slovakia was one of the countries experiencing a high burden of COVID-19 infections, and hospitals faced greater challenges in preventing and managing CLABSI; (2) Methods: A retrospective analysis of CLABSI data from all patients admitted to adult respiratory intensive care units before and during COVID-19 pandemic was conducted. We followed the guidelines of the Center for Disease Control surveillance methodology for CLABSI. Data were analyzed using STATISTICA 13.1; (3) Results: We analyzed the data of 803 ICU patients hospitalized for 8385 bed days, with 7803 central line days. Forty-five CLABSI events were identified. The CLABSI rate significantly increased during the COVID-19 pandemic compared to before the COVID-19 pandemic (2.81 versus 7.47 events per 1000 central line days, (*p* < 0.001). The most frequently identified pathogens causing CLABSI were Gram-negative organisms (60.20%). The risk factors found to increase the probability of developing CLABSI were length of stay (OR = 1.080; 95% Cl: 1.057–1.103; *p* < 0.001) and COVID-19 (OR = 5.485; 95% Cl: 32.706–11.116; *p* < 0.001). (4) Conclusions: The COVID-19 pandemic was associated with increases in CLABSI in ICUs. These data underscore the need to increase efforts in providing surveillance of CLABSI and implementing infection prevention measures.

## 1. Introduction

Healthcare-associated infections (HAI), especially device-associated infections (DAI), have been a significant concern in healthcare settings for many years [1,2]. Central line-associated bloodstream infections (CLABSI) are one of the most important DAI. CLABSI is associated with extra mortality, and extra costs, and represents one of the most serious challenges to patient safety and quality healthcare in intensive care unit (ICU) settings. Nevertheless, these infections are highly preventable [3]. Various strategies, such as implementing evidence-based guidelines, improving hand hygiene, using antimicrobial-impregnated catheters, and enhancing healthcare provider education and training, have been employed to prevent and reduce CLABSI [4]. In many hospitals, CLABSI rates had been steadily declining before the COVID-19 pandemic [5,6].

COVID-19, caused by the severe acute respiratory syndrome coronavirus 2 (SARS-CoV-2 virus), primarily affects the respiratory system. However, its impact extends beyond respiratory health and has implications for various aspects of healthcare, including the incidence of CLABSI [7]. Several specific factors can have an impact on increasing the rate of HAI. People with COVID-19 often require admission to the ICU [8]. Critically ill patients, including those with COVID-19, frequently need central venous catheters (CVC) for various treatments, which may contribute to a higher risk of CLABSI. Multiple studies have shown that ICU patients with COVID-19, like those with other respiratory viruses, are more susceptible to developing bacterial co-infections, including bloodstream infections [9,10]. Infection prevention and control (IPC) proGrams were mainly focused on pandemic management during the COVID-19 pandemic [11].

Many studies showed a significant increase in CLABSI rates during the COVID-19 pandemic [12,13,14]. Other studies did not find a significant impact of the COVID-19 pandemic on the rates of HAI [15,16]. Some other studies showed a decrease in CLABSI, but without statistical significance [17,18]. The specific impact of COVID-19 on CLABSI may vary across healthcare settings and regions. Slovakia was one of the countries experiencing a high burden of COVID-19 infections and hospitalizations and faced greater challenges in preventing and managing CLABSI due to the increasing pressure on healthcare resources and overcrowding [19].

There is a lack of national data determining the relevant number of HAI or DAI in Slovakia. The HAI data are underestimated, and IPC proGrams cannot be implemented properly. According to our knowledge, there is no previously published data about the rate of CLABSI in ICU settings during the COVID-19 pandemic in Slovakia. The aim of this study was to assess the incidence, pathogens, and risk factors of CLABSI and to estimate the impact of the COVID-19 pandemic on the rates of CLABSI.

## 2. Materials and Methods

A retrospective analysis of prospectively collected data on central line-associated infections was conducted. The study was performed in a 400-bed teaching hospital in Slovakia over a period of 4 years, before the COVID-19 pandemic (January 2017–November 2019) and during the COVID-19 pandemic (October 2020–August 2022). The surveillance study included all patients admitted to adult respiratory intensive care units.

The data obtained before the COVID-19 pandemic were compared with the data during the pandemic. The comparison of the data was also conducted from patients hospitalized in ICU both with and without COVID-19 disease during the COVID-19 pandemic. Data from patient admission to patient discharge, regarding the patient’s age, gender, hospitalization type (medical or surgical), number of bed days, central line days, CLABSI rates and cultures, ICU outcome (dead or alive), and COVID-19 disease were obtained.

The study was performed according to the guidelines of the Center for Disease Control surveillance methodology for CLABSI, which allows benchmarking. The criteria of Centre for Disease Control and Prevention’s National Healthcare Safety Network (CDC-NHSN) were used for CLABSI definitions and denominators [20].

According to these criteria, CLABSI is a type of healthcare-associated infection that occurs when a central venous catheter is in place within the previous 48 h or more before the onset of the infection, and there is evidence of bloodstream infection that is not related to an infection at another site. Laboratory-confirmed bloodstream infection: This means that the patient must have at least one positive blood culture from a recognized pathogen or two or more positive blood cultures from common skin contaminant organisms (such as *Staph. epidermidis*). The positive blood cultures must be collected on separate occasions and be related to the infection’s signs and symptoms.

Central line utilization ratio (CL-ratio) was calculated by comparing the total number of days a central line was in use to the total number of patient days during a specific time period. The CL-ratio helps healthcare facilities monitor the appropriateness and necessity of using central lines and identify potential opportunities for improvement in central line use.

Central line-associated bloodstream infection (CLABSI) per 1000 central line days was calculated by dividing the total number of CLABSI cases during a specific time period by the total number of central line days during the same period and multiplying by 1000 (CLABSI rate per 1000 central line days = (Number of CLABSI cases/Total number of central line days) × 1000). Calculating the CLABSI rate per 1000 central line days allows healthcare facilities to track infection rates and assess the effectiveness of infection prevention measures over time. It is a valuable metric for monitoring patient safety and the quality of care provided.

All COVID-19 patients had laboratory-confirmed SARS-CoV-2 infection using real-time polymerase chain reaction (PCR).

The identification of microorganisms was performed using biochemical tests ENTEROtest 16 (Erba Lachema Ltd., Brno, Czech Republic) and API 10S test (Biomérieux Ltd., Prague, Czech Republic).

The final form of the logistic regression model with qualitative explanatory variables identified determinants contributing to the occurrence of CLABSI among patients under observation. The verification and construction of the logistic model was based on Wald’s statistics. The goodness of fit of the model was verified using the Hosmer–Lemeshow test.

Additionally, an ROC curve was constructed to assess the compliance of the CLABSI occurrence indications resulting from the model with the actual indications. The area under the ROC curve was calculated (denoted as AUC—area under the curve), which is a measure of the goodness of the model.

For comparison of distribution between the analyzed quantitative variables concerning the occurrence of CLABSI, comparison before and during the pandemic, and the occurrence or absence of COVID-19 disease, a non-parametric U Mann-Whitney test was used. The χ^2^ test was employed to test the independence between the two qualitative variables.

Configuration Frequency Analysis (CFA) was performed to determine the actual observed frequencies of a given combination of variables (dichotomous variables) related to the occurrence of CLABSI, COVID-19, the COVID-19 pandemic, and death of patients. Expected values were used, representing the average expected frequencies of a given combination of variables from the values of standardized statistics normal distribution. A *p*-statistic value was given for a statistic indicating whether a given combination of variables is significantly greater or smaller than the expected value.

In all statistical tests, a significance level of α = 0.05 was assumed. Statistical analysis was performed using the advanced analytical software STATISTICA version for Windows 13.1 TIBCO Software Inc.—StatSoft, Poland.

## 3. Results

During the two study periods, we analyzed the data of 803 intensive care unit patients hospitalized for 8385 bed days, with 7803 central line (CL) days. A total of 339 patients were hospitalized for 3098 bed days in the respiratory ICU before the COVID-19 period (January 2017–November 2019), and 464 patients were hospitalized for 5299 bed days during the COVID-19 pandemic (October 2020–August 2022), out of which 207 patients were diagnosed with COVID-19 and were hospitalized in the COVID-19 ICU. There was a significant 25% increase in length of stay (LOS) during the COVID-19 pandemic. The average LOS was 9.10 before and 11.42 during the COVID-19 pandemic, (*p* < 0.001). Additionally, a significant 45% increase in LOS (13.80) was observed in patients with COVID-19 compared to the LOS (9.50) of patients without COVID-19 during the COVID-19 pandemic, (*p* < 0.001). Detailed data on the CLABSI rates and characteristics before and during the COVID-19 pandemic are presented in Table 1.

Of the total 45 (5.6%) CLABSI events, 8 (2.36%) occurred before the COVID-19 pandemic, while 37 (7.97%) occurred during the COVID-19 pandemic, (*p* < 0.001) (see Table 1). Before the COVID-19 pandemic, 2848 CL days were reported, while during the COVID-19 pandemic 4955 CL days were reported. CLABSI rates significantly increase by 166% from 2.81 to 7.47 events per 1000 central line days, (*p* < 0.001).

There was also a significant difference (136% increase) in CLABSI rates between patients without COVID-19 and patients with COVID-19 (4.33 versus 10.20 events per 1000 central line days) in ICUs during the COVID-19 pandemic (*p* < 0.001). Compared to before the pandemic, CLABSI rates increased by 54% during the pandemic in patients without COVID-19 disease (2.81 versus 4.33 events per 1000 central line days). We found a significant increase in the central line utilization ratio from 0.92 before the COVID-19 pandemic to 0.94 during the COVID-19 pandemic (*p* < 0.001).

Comparing patients with and without CLABSI to identify confounders, we found that patients with CLABSI were younger, (*p* = 0.032). Significant factors influencing the acquisition of CLABSI were medical type of hospitalization, coronavirus disease, and the COVID-19 pandemic (*p* < 0.001). In contrast, no significant differences were found between gender (*p* < 0.715) and mortality (*p* < 0.138). The analysis of confounding variables is shown in Table 2.

The frequency of occurrence of the type of microorganisms before and during the COVID-19 pandemic was not statistically significantly different (*p* = 0.258). However, the most frequently isolated pathogens causing CLABSI were Gram-negative bacteria (60.20%), followed by Gram-positive bacteria and fungi. The main CLABSI pathogens were *Acinetobacter* spp. (22.22%), *Klebsiella pneumoniae* (20.20%), *Pseudomonas aeruginosa* (17.78%), *Enterococci* spp. (13.33%), coagulase-negative *Staphylococci epidermidis* (11.11%) *Streptococcus* spp. (6.67%), *Escherichia coli* (4.44%), *Enterobacter cloacae* (2.22%) and *Candida albicans* (2.22%). Detailed data on the frequency of microorganisms detected in positive CLABSI are presented in Table 3.

Based on the estimated logistic regression, the chance to acquire CLABSI is 5.5 times greater in patients with COVID-19, compared to the patients without COVID-19 (OR = 5.485; 95% Cl: 32.706–11.116; *p* < 0.001). The chance of CLABSI increases with the length of hospital stay (OR = 1.080; 95% Cl: 1.057–1.103; *p* < 0.001) and the younger age of the patient (OR = 0.971; 95% Cl: 0.949–0.993; *p* = 0.011). Detailed data on logistic regression of predictors of CLABSI are presented in Table 4.

The value of the statistic Hosmer–Lemeshow = 9.401, with a *p*-value of 0.309, indicates a significant model fit of the logistic regression model. Based on the analysis of the area under the ROC curve, one can also state that the model fits the data well (the area under the curve is AUC = 0.889) (Figure 1) and is characterized by a good predictive ability resulting from the obtained sensitivity.

In Table 5, data are presented to determine whether the given combination of variables is significantly greater or smaller than the expected value. The main results show that during the COVID-19 pandemic, 14 patients died, and 13 patients survived when they had both CLABSI and COVID-19. Additionally, 111 patients died because of COVID-19 disease without CLABSI, and 60 patients died without a diagnosed COVID-19 or CLABSI. Before the COVID-19 pandemic, 5 patients survived when they had CLABSI.

## 4. Discussion

The data on healthcare-associated infections (HAI) from the national reporting system are underestimated in Slovakia. Many healthcare settings still fail to achieve the basic goal of reporting the real HAI data, with some denying the existence of these infections unprofessionally. Some healthcare settings have reported no HAI infections [21]. Only a few hospitals are involved in the surveillance and reporting of HAI in intensive care units (ICU) according to the European Center for Infection Prevention and Control (ECDC) protocol, providing more accurate data [21,22]. However, due to the COVID-19 pandemic, data collection and reporting to the ECDC were not carried out in ICUs in Slovakia [21].

We decided to evaluate data on central line-associated bloodstream infection (CLABSI) incidence because CLABSI is an important marker of quality in hospitals. To the best of our knowledge, this is the first study in Slovakia comparing CLABSI rate, etiology, and risk factors in critically ill patients before and during the COVID-19 pandemic.

Studies have shown that before the COVID-19 pandemic, decreasing CLABSI incidence rates had been observed, where surveillance and prevention control were performed [5,6]. Other studies did not find a difference in CLABSI rates. For example, the ECDC point prevalence study (PPS) of HAI performed in the European Union/European Economic Area (EU/EEA) in 2016–2017 revealed a 2.1% prevalence of CLABSI and the ECDC concluded that there was no major overall progress seen in preventing healthcare-associated infections compared to ECDC PPS results from 2011 to 2012 [22].

According to our results, before the COVID-19 pandemic, the incidence of CLABSI was 2.36%. Our CLABSI rates were higher (2.81 per 1000 central line days) before the COVID-19 pandemic compared to pooled data reported from CDC-NHSN ICUs (0.8 per 1000 central line days) [23], but lower than the data reported by the International Nosocomial Infection Control Consortium (INICC) (5.30 per 1000 central line days) [24]. This indicates that even before the COVID-19 pandemic, we had a deficiency in CLABSI surveillance, prevention and control. CLABSI is highly preventable, and when countries invest more in infection prevention and control, this is reflected at different levels of the prevention process, including HAI rate reduction.

Our study highlights a higher rate of CLABSI during the COVID-19 surge compared with the pre-pandemic period in ICU patients. Our findings are consistent with previous studies conducted in many healthcare settings [12,13,25,26,27]. For example, analyses conducted in US NHSN hospitals showed 37–51% increases in CLABSI rates during the COVID-19 pandemic. According to their and other results, larger hospitals (>300 beds), as well as ours, were most affected [13,25]. Comparable studies show an increase in CLABSI rate by 280% in tertiary care hospitals in Detroit [28], a 60% increase in HCA Healthcare-affiliated hospitals [25], an 86% in INNIC developing countries [26], and a 16% increase were reported in Saudi Arabia [12]. In our study, we also analyzed CLABSI rates in patients with and without COVID-19 during the COVID-19 pandemic. During the COVID-19 pandemic, patients hospitalized for COVID-19 had a significantly higher CLABSI rate (136% increase) than patients without COVID-19 (4.33 versus 10.20 per 1000 central line days). The study conducted in a tertiary care teaching hospital also confirms a significantly higher CLABSI rate in the COVID-19 ICU compared to the non-COVID-19 ICU. After implementing quality improvement measures in the hospital, they were able to reduce the rate of non-COVID-19 CLABSI, but the rate of COVID-19 CLABSI remained unchanged [29].

We have already mentioned a sharp 166% increase in CLABSI rate compared to before and during the COVID-19 pandemic. Reasons for this increase can be attributed to several factors that were influenced by the unique circumstances of the pandemic. Slovakia was hard hit by the COVID-19 epidemic mainly at the end of 2020 and in early 2021, when the number of new cases rapidly increased, and patients at risk of dying began to exceed the capacities of hospitals [19]. Based on our findings, CLABSI rates were consistent with the worsening COVID-19 epidemiological situation, and the same results were presented in other studies [25,30].

The key reasons for the CLABSI increase are process-related and patient-specific [29]. During the COVID-19 pandemic, hospitals faced significant strain in terms of staff, equipment, and supplies. Due to the rapid increase in COVID-19 cases, our but also other hospitals needed to quickly train and redeploy staff to assist in the ICU care of COVID-19 patients [31]. Additionally, healthcare settings had staffing shortages due to exposure to the SARS-CoV-2 virus, illness, or the need to care for family members. Healthcare workers were under immense pressure during the pandemic, working long hours and dealing with high patient loads in critical conditions. These factors, combined with the extensive use of personal protective equipment (PPE), may have led to PPE fatigue and lapses in adherence to infection control protocols, including proper central line insertion and maintenance techniques. This also caused a substantial number of nurses to express concerns about their working conditions due to heavy workloads and inadequate salaries, leading to a considerable proportion of them contemplating leaving their positions during the COVID-19 pandemic in Slovakia [32].

Infection control personnel experienced an increase in consultations during the pandemic, which primarily focused on COVID-19 isolations and exposures, and some routine healthcare activities, including infection prevention practices targeting CLABSI, may have been disrupted or deprioritized [33]. According to the Infectious Disease International Research Initiative (ID-IRI) survey conducted in 2021, which analyzed CLABSI prevention bundles in 22 countries, 17.4% of hospitals had no surveillance system for CLABSI, 7.1% of ICUs had no CLABSI bundle and 23% had no dedicated checklist. The study demonstrated significant differences in central line bundles between low/low-middle income countries with higher and middle-higher income [34].

Moreover, COVID-19 patients had a higher risk for CLABSI and death [35]. COVID-19 patients in ICUs were associated with longer hospitalization, multiple comorbidities and longer usage of invasive devices [13,28]. COVID-19 patients, especially those severely ill and requiring ICU care, are often immunosuppressed or have weakened immune systems. Using immunosuppressive drugs among COVID-19 patients increased their susceptibility to infections [36]. According to our results, we confirmed that CLABSI was associated with COVID-19 diagnosis, longer hospitalization, and lower age. Younger patients had greater a chance of CLABSI. This could be a result of the lower age of patients who needed to be hospitalized in ICU settings during the COVID-19 pandemic compared to the age of patients before the COVID-19 pandemic. The CLABSI rates were not influenced by gender or central line days because they were similar in observed periods. These are similar findings to the study conducted by the International Nosocomial Infection Control Consortium (INICC) [20]. We did not find a significant association between CLABSI and mortality. Higher influence on mortality had COVID-19 disease.

The presence and prevalence of specific microorganisms causing CLABSI during the COVID-19 pandemic can be influenced by factors such as the patient population, healthcare practices, and the level of infection control measures implemented during the pandemic. COVID-19 patients have shown co-infection rates from 7% to 15%, and 27% of those who die were co-infected. However, it appears that nosocomial origins for co-infection might be a major factor [35]. These microorganisms are often multidrug resistant. Patients with CLABSI caused by multidrug-resistant organisms are often more challenging to treat, leading to higher rates of complication, prolonged hospital stays, mortality, increased healthcare costs and spreading of resistance [30]. In our study, the most frequently isolated pathogens causing CLABSI were Gram-negative organisms (60.20%) and there was no difference between our study periods. Similar results to ours have been confirmed in other studies in which also Gram-negative microorganisms were isolated from CLABSI cultures [25,37] and *Acinetobacter* was the most frequent [37]. Some other studies showed that the CLABSI were caused predominantly by Gram-positive microorganism [13] or by *Candida* spp. during the COVI-19 pandemic [13,38]. *Acinetobacter* spp., *Klebsiella pneumoniae*, and *Pseudomonas aeruginosa*, which were mostly detected in blood cultures of our patients, are known to persist for long periods in the hospital environment [39,40,41] and can be transmitted through contact with contaminated surfaces of healthcare professionals´ hands. It is necessary to increase awareness of and compliance with standard hygiene precautions to limit the spread of microorganisms in an intensive care unit as much as possible [42].

The main strength of this study is its ability to collect and compare CLABSI data before and during the COVID-19 pandemic in ICU patients, as there is a lack of these data in Slovakia. However, we also can recognize the limitations of our study. To be able to generalize our findings on a national level, more healthcare settings should be analyzed, but we believe our data will serve as the baseline for obtaining valid data on CLABSI in other ICU settings.

## 5. Conclusions

The COVID-19 pandemic was associated with increased CLABSI rates in ICUs. CLABSI was mostly caused by Gram-negative microorganisms. CLABSI did not have a direct impact on mortality, but the occurrence of COVID-19 and prolonged hospitalization did. Our results underline the need to increase the efforts in providing surveillance of CLABSI and ensuring infection prevention through close monitoring of processes and outcomes related to device use, as well as regular feedback on performance to the ICU staff and clinical leaders. Further studies are needed to better understand the relationship between COVID-19 and the increased rates of CLABSI.

## Figures and Tables

**Figure 1 healthcare-11-02415-f001:**
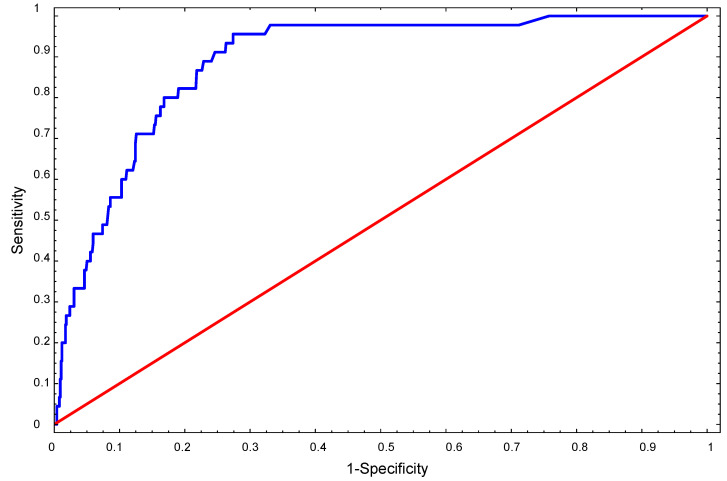
The ROC curve graph–Sensitivity and Specificity of logistic regression model. ---- Area under the ROC; ---- Random clasiffier.

**Table 1 healthcare-11-02415-t001:** Central line associated infection (CLABSI) rates and characteristics before and during the COVID-19 pandemic.

Variable	Total (n = 803)	Before Pandemic (n = 339)	Pandemic (n = 464)	*p*-Value	Pandemic		*p*-Value
					Yes COVID-19 (n = 207)	Non COVID-19 (n = 257)	
No. of CLABSI n (%)	45 (5.60)	8 (2.36)	37 (7.97)	<0.001 *^B^	27 (13.04)	10 (3.09)	<0.001 *^B^
No. of Bed Days (M, SD, 95% CI)	8385 (10.44 ± 11.22; 9.665–11.219)	3098 (9.10 ± 11.11; 7.916–10.290)	5299 (11.42 ± 11.20; 10.398–12.442)	<0.001 *^A^	2857 (13.80 ± 11.11; 12.279–15.325)	2442 (9.50 ± 10.93; 8.160–10.844)	<0.001 *^A^
Central line days (M, SD, 95% CI)	7803 (9.72 ± 12.55; 8.848–10.587)	2848 (8.40 ± 12.72; 7.038–9.765)	4955 (10.68 ± 12.32; 9.554–11.803)	<0.001 *^A^	2647 (12.79 ± 11.67; 11.188–14.387)	2308 (8.98 ± 12.59; 7.433–10.528)	<0.001 *^A^
Central line utilization ratio (95% CI)	0.93 (0.925–0.935)	0.92 (0.909–0.928)	0.94 (0.928–0.941)	<0.001 *^A^	0.93 (0.916–0.936)	0.95 (0.935–0.953)	0.001 *^A^
CLABSI/1000 CL-Days (95% CI)	5.77 (4.313–7.707)	2.81 (1.424–5.533)	7.47 (5.423–10.270)	<0.001 *^A^	10.20 (95 CL 7.02–14.8)	4.33 (2.355–7.957)	<0.001 *^A^

Note. CI = confidence interval, CL = central line; CLABSI = central line-associated bloodstream infection; COVID-19 = coronavirus disease; M = mean; SD = standard deviation; ^A^ Mann–Whitney Test, ^B^ χ^2^ test, * *p* < α; α = 0.05 statistical significance found.

**Table 2 healthcare-11-02415-t002:** Analysis of confounding variables.

Variable	Variants	Total (n = 803)	CLABSI No (n = 758)	CLABSI Yes (n = 45)	*p*-Value
Age Me (IQR)	Years	64 (21.50)	64 (21.37)	59 (17)	0.032 **^A^
Gender n (%)	0 Male	515 (64.13)	485 (63.98)	30 (66.67)	0.715 ^B^
	1 Female	288 (35.87)	273 (36.02)	15 (33.33)	
Hospitalization Type n (%)	0 Surgical	371 (46.20)	364 (48.02)	7 (15.56)	<0.001 **^B^
	1 Medical	432 (53.80)	394 (51.98)	38 (74.39)	
ICU Outcome n (%)	No	528 (65.75)	503 (66.36)	25 (55.56)	0.138 ^B^
	Yes	275 (34.25)	255 (33.64)	20 (44.44)	
COVID-19 n (%)	No	596 (74.22)	578 (76.25)	18 (40.00)	<0.001 **^B^
	Yes	207 (25.78)	180 (23.75)	27 (60.00)	
Pandemic COVID-19 n (%)	No	339 (42.22)	331 (43.67)	8 (17.78)	<0.001 **^B^
	Yes	464 (57.78)	427 (56.33)	37 (82.22)	

Note. CLABSI = central line-associated blood stream infection; COVID-19 = coronavirus disease; ICU = intensive care unit; IQR = interquartile range; Me = mean; ^A^ Mann-Whitney Test, ^B^ χ^2^ test, ** *p* < α; α = 0.05 statistical significance found.

**Table 3 healthcare-11-02415-t003:** Frequency of microorganisms isolated in positive blood cultures in hospitalized patients before the COVID-19 pandemic and during the COVID-19 pandemic.

Microorganism CLABSI	Total N = 45 (%)	Pandemic COVID-19 Non 8 (%)	Pandemic COVID-19 Yes 37 (%)
*Acinetobacter* spp.	10 (22.22)	2 (25.00)	8 (21.62)
*Candida albicans*	1 (2.22)	1 (12.50)	-
Coagulase-negative *Staphylococci epidermidis*	5 (11.11)	1 (12.50)	4 (10.81)
*Enterobacter cloacae*	1 (2.22)	-	1 (2.70)
*Enterococci* spp.	6 (13.33)	-	6 (16.22)
*Escherichia coli*	2 (4.44)	-	2 (5.41)
*Klebsiella pneumoniae*	9 (20.20)	1 (12.50)	8 (21.62)
*Pseudomonas aeruginosa*	8 (17.78)	3 (37.50)	5 (13.51)
*Streptococcus* spp.	3 (6.67)	-	3 (8.81)

Note. CLABSI = central line-associated blood stream infection; COVID-19 = coronavirus disease; spp. = species (plural).

**Table 4 healthcare-11-02415-t004:** Risk predictors of central line associated infection (CLABSI).

Variable–Reference Variant	Estimate of the Logistic Regression Parameter	OR (95% Cl)	*p*-Value
Constant term	−3.058	0.047 (0.012–0.182)	<0.001
Patients’ days	0.077	1.080 (1.057–1.103)	<0.001
COVID-19	1.702	5.485 (2.706–11.116)	<0.001
Age	−0.029	0.971 (0.949–0.993)	0.011

Note. CLABSI = central line-associated bloodstream infection; COVID-19 = coronavirus disease.

**Table 5 healthcare-11-02415-t005:** Frequencies of a combination of central line-associated infection (CLABSI), the COVID-19, pandemic, COVID-19, and intensive care unit (ICU) outcome. Actual observed frequencies of a given combination of variables.

CLABSI	COVID-19	Pandemic COVID-19	ICU Outcome	Observed Values No. of Patients	Expected Values No. of Patients	χ^2^	*p*-Value
Non	Non	Non	Non	247.00	156.17	7.268	0.000
Non	Non	Non	Yes	84.00	81.34	0.295	0.384
Non	Non	Yes	Non	187.00	213.76	1.830	0.034
Non	Non	Yes	Yes	60.00	111.33	4.865	0.000
Non	Yes	Yes	Non	69.00	74,.4	0.608	0.272
Non	Yes	Yes	Yes	111.00	38.67	11.632	0.000
Yes	Non	Non	Non	5.00	9.27	1.403	0.080
Yes	Non	Non	Yes	3.00	4.83	0.832	0.203
Yes	Non	Yes	Non	7.00	12.69	1.597	0.055
Yes	Non	Yes	Yes	3.00	6.61	1.404	0.080
Yes	Yes	Yes	Non	13.00	4.41	4.093	0.000
Yes	Yes	Yes	Yes	14.00	2.30	7.725	0.000

Note. CLABSI = central line-associated bloodstream infection; COVID-19 = coronavirus disease; ICU = intensive care unit. Expected: Mean expected frequencies of a given combination of variables. *p*: *p*-value for the z statistic Type/Antitype. z: Standardized Normal Distribution statistic value.

## Data Availability

Data are available upon request from the corresponding author.

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
