# Peer review of "Central Line Associated Bloodstream Infections in Critical Ill Patients during and before the COVID-19 Pandemic"

_healthcare, 2023, doi:10.3390/healthcare11172415_

Round 1
Reviewer 1 Report
In this report, authors evaluated Central Line Associated Bloodstream Infections in Critical Ill Patients During and Before the Covid-19 Pandemic in Slovak Republic. This is an important topic, and this is a well-designed and written article.
I have some minor points and recommendations.
1- In the Abstract section, Background part, authors need to add why they planned to perform this study. Original version includes the aim of this study.
2- Authors prefer to evaluated time period: before the COVID-19 pandemic (January 2017 - November 2019) 75 and during the COVID-19 pandemic (October 2020 – August 2022). Why you exclude January-October 2020 (the first phase of pandemic, majority of the patients followed-up in the hospital, isolation period is longer (sometimes more than 10 days) etc.
3- The numbers of CLABSI rate is lower than expected regarding the bed capacity of the hospital. Only 45 cases with CLABSI. Do you have a chance to calculate SIRs (standardized infection ratios) for CLABSI before and after pandemic with observed and expected events. Al Saffar and colleagues recent article is a good example for this purpose (study is not related with me) (https://www.ncbi.nlm.nih.gov/pmc/articles/PMC10072974/)
4- Authors make comparison between patients with COVID 19 and patients hospitalized in ICU without COVID-19. Do you think that there are changes for patients age, disease, prognosis profile (for non COVID patients). What is the patient’s profile for adult respiratory intensive care units (only case with respiratory disorders, or including trauma patients, post-surgical cases?)
5- In the Results section, authors mentioned that there is statistically significant difference for central line utilization rate (0.93 (0.925- 0.935) vs. 0,92 (0.909- 0.928) 0.94 (0.928- 0.941); p<0.001. Very strong difference but the ratios are quite similar. Need to check.
6- For confounding variable, age is statistically significant however CLABSI positive patients were younger than negative ones (not expected situation regarding previous studies).
7- In Table 2, what is the difference between COVID-19 and pandemic COVID-19?
8- I prefer to learn the Author’s comments about potential effect of mitigation strategies during the COVID19 pandemic on HAI. As authors mentioned, we well known that improving hand hygiene, and enhancing healthcare provider education and training, have been employed to prevent and reduce CLABSI. During the pandemic period, all health care providers wear a mask, prefer to handwashing more common than the pre-pandemic period, low community acquired infection, low antibiotic use in the community and also no visitors in the hospital setting. In contrast, we well know that the work load of HCWs increased during the pandemic, increased number of patients in the ICU due to COVID. Do you think that this extraordinary condition affects the infection rate in the hospital?
Reviewer 2 Report
Sona Hliknova with co-authors wrote an article entitled “Central Line Associated Bloodstream Infections in Critical Ill 2 Patients During and Before the Covid-19 Pandemic”.
I find this topic important and timely. The Authors pointed out that there is a lack of national data determining the relevant number of HAI or DAI in Slovakia (lines 65-66), which increases additionally the importance of the presented data.
The study design is described clearly and reasonably. The fact that results obtained by the Authors are consistent with other studies is important and increases the impact of the presented data.
I do not work in the ICU and I believe that is the reason why I do not see an increase of bloodstream infection caused by Acinetobacter spp, as this is bacteria that was rarely found in my patients. I do realize it is the difference between ICUs and non-ICUs. Nevertheless, I also see an increase in bloodstream infection occurrence and I agree that Gram-negative bacteria can be found in blood cultures more often than before the COVID-19 pandemic.
There is one thing that might be missing in the article. Together with my colleagues, we wonder if the reason for higher bloodstream infection occurrence is not extensive antibiotic use during COVID-19 infection, especially since we see an increase in antibiotic resistance in bacteria found in blood cultures. Did the Authors analyze if the increase in bloodstream infection was related to previous extensive antibiotic use in patients? I also realize that the patient who requires ICU had pulmonary failure and required mechanical ventilation, thus, that patient had been hospitalized before he was admitted to ICU. Did the Authors have data such as the number of hospitalizations before admission to the ICU or the length of hospitalization before admission to the ICU?
Reviewer 3 Report
Thank you for allowing me to review this excellent article. It has good scientific merit and will be interesting to a wide variety of readers, including microbiologists.
Criteria for bloodstream infection could be refined further. Specifically, if a skin contaminant (such as Staph. epidermidis) is isolated, the same contaminant must be isolated on at least two consecutive occasions.
The method of bacterial identification (e.g. 16S rRNA sequencing or MALDI-TOF MS) should be stated.
The “95 CIs” in Table 1 should more accurately be “95% CI.”
There is no need to italicize “spp.” because this is not a specific species name but rather an abbreviation.
In the discussion, were some decimal points displayed as commas? This contrasts with the way data were presented in the results, where decimal points were displayed as periods. For instance, is “2,1%” on line 250 supposed to be “2.1%?”
The English is easy to understand. However it is not perfect and some points could be rephrased to sound better. For instance, on line 185, "statistically insignificantly different" could be refined although it is still understandable nonetheless.
Round 2
Reviewer 1 Report
Thank you for your responses and corrections.